# Reverse-ChIP Techniques for Identifying Locus-Specific Proteomes: A Key Tool in Unlocking the Cancer Regulome

**DOI:** 10.3390/cells12141860

**Published:** 2023-07-14

**Authors:** Tim M. G. MacKenzie, Rocío Cisneros, Rajan D. Maynard, Michael P. Snyder

**Affiliations:** 1Genetics Department, Stanford University, Stanford, CA 94305, USA; tmackenz@stanford.edu (T.M.G.M.);; 2Sarafan ChEM-H/IMA Postbaccalaureate Fellow in Target Discovery, Stanford University, Stanford, CA 94305, USA

**Keywords:** chromatin architecture, mass spectrometry proteomics, quantitative mass spectrometry, locus-specific chromatin isolation, proximity labeling, genome regulation, transcriptional regulation, promoter pulldown proteomics, telomeres, cancer regulome

## Abstract

A phenotypic hallmark of cancer is aberrant transcriptional regulation. Transcriptional regulation is controlled by a complicated array of molecular factors, including the presence of transcription factors, the deposition of histone post-translational modifications, and long-range DNA interactions. Determining the molecular identity and function of these various factors is necessary to understand specific aspects of cancer biology and reveal potential therapeutic targets. Regulation of the genome by specific factors is typically studied using chromatin immunoprecipitation followed by sequencing (ChIP-Seq) that identifies genome-wide binding interactions through the use of factor-specific antibodies. A long-standing goal in many laboratories has been the development of a ‘reverse-ChIP’ approach to identify unknown binding partners at loci of interest. A variety of strategies have been employed to enable the selective biochemical purification of sequence-defined chromatin regions, including single-copy loci, and the subsequent analytical detection of associated proteins. This review covers mass spectrometry techniques that enable quantitative proteomics before providing a survey of approaches toward the development of strategies for the purification of sequence-specific chromatin as a ‘reverse-ChIP’ technique. A fully realized reverse-ChIP technique holds great potential for identifying cancer-specific targets and the development of personalized therapeutic regimens.

## 1. Introduction

Among the many remarkable phenotypes differentiating cancer from healthy states, transcriptional dysregulation is one of the most marked [1]. Indeed, targeted cancer therapies based on tumor-specific transcriptional signatures were predicted at the dawn of the human genome era and have continued to receive attention as a promising strategy [2,3]. Integrated analysis of genomic, transcriptomic, and epigenomic data across multiple lung cancer cell lines identified unique and shared transcriptional signatures that led the authors to posit the existence of cancer ‘regulome’ [4]. Aberrant regulation of cellular pathways can lead to specific cancer types from diverse underlying molecular etiologies [5,6].

Transcriptional regulation is determined by both *cis*- and *trans*-acting factors. Promoters proximal to gene bodies and enhancers distal from gene bodies are *cis* elements that contain recognition elements for transcription factors to bind to sequence-specific elements. Transcription factors in turn recruit other *trans*-acting factors involved in transcription and chromatin remodeling. Chromatin remodelers can install epigenetic marks, such as histone posttranslational modifications and DNA methylation, altering the standard ‘histone code’ [7] into cancer-associated epigenomic signatures [8,9]. These altered chromatin states can impact cellular phenotype in the cancer state in surprising ways [10]. Identification of transcription factors and chromatin remodelers determining the epigenetic state of cancer-associated gene promoters can provide a list of candidate therapeutic targets.

A key experimental technique in the advancement of our understanding of transcriptional regulation was the development of chromatin immunoprecipitation (ChIP) (Figure 1a). Antibodies purify both the target antigen and its associated chromatin, including protein complexes and DNA sequences, from the cellular milieu. The tight association between positively charged nucleosomes and the negatively charged phosphate backbone of DNA enables selective purification of chromatin using histone antibodies [11], while the addition of a formaldehyde cross-linking step [12] enables the detection of nonhistone proteins as well [13]. The chemistry and biochemistry of formaldehyde cross-linking have been excellently reviewed elsewhere [14]. ChIP is invaluable in forward genetic studies, enabling researchers to identify the downstream targets of disease-relevant factors.

Detection of specific protein-DNA interactions was initially assessed through Southern blot hybridization or PCR-based amplification of a target sequence that was hypothesized to interact with the purified chromatin component. The development of DNA microarrays massively expanded the ability to simultaneously probe multiple sequences, substantially increasing the fraction of the genome that could be studied in an individual experiment [15,16]. ChIP went truly genome-wide for human studies when it was combined with next-generation sequencing (ChIP-Seq) [17,18,19,20]. In this method, the only requirement for determining all interactions of a factor of interest across the genome is a specific antibody; experiments can be easily adapted to compare different conditions to understand transcriptional dynamics in response to specific experimental challenges. As the technique has matured, a focus of ChIP-Seq research has been the development of bioinformatics strategies to analyze the genomic-scale data collected in every experiment [21]. Novel experimental strategies, such as CUT&RUN [22] and CUT&Tag [23], have addressed bias and background issues found in standard ChIP-Seq experiments, moved towards single base-pair resolution, and enabled the use of much smaller sample input to produce reliable data. The development and application of ChIP-Seq and related techniques to study transcriptional regulation have been reviewed elsewhere [24,25,26,27,28].

Despite the power of ChIP-Seq to uncover structural and functional elements of the genome, there are limitations that cannot be overcome virtue by the experimental design. The reliance on high-quality antibodies limits what factors can be studied, preventing the discovery of novel genome regulators. Though the genome-wide binding of a given factor is uncovered, the context of binding in any particular genomic locus is missing; any cofactors that bind a specific genomic locus remain undiscovered. Affinity purification of a factor known to bind to a specific locus followed by mass spectrometry cannot determine a locus-specific proteome. The strength of ChIP-Seq, finding genome-wide binding of a specific factor, turns into a liability when attempting to construct a locus-specific proteome. Serial ChIP-Seq experiments are required to build an understanding of the regulatory landscape in a region of interest, requiring a specific hypothesis of potential factors due to the targeted nature of the method. As a complementary experimental approach, many labs have worked toward the development of reverse-ChIP strategies to enable the unbiased assessment of regulatory factors at any arbitrarily chosen genomic locus of interest (Figure 1b) [29].

A reliable reverse-ChIP approach would prove as powerful for reverse genomics as ChIP-Seq has been for forward genomics. Understanding how genomic differences give rise to phenotypic diversity requires identifying the molecular mediators that are present in experimentally identified regions. An ideal reverse-ChIP technique would involve the enrichment of an experimenter-defined genomic region to homogeneity from background chromatin followed by identification of the molecular composition of the region. Identifying the molecular composition of an isolated locus would involve determining the presence of protein and nucleic acid factors, their relative stoichiometry, and any associated posttranslational or posttranscriptional modifications. Achieving this ambitious goal is limited in practice by experimental realities. For example, histone proteins are ubiquitously present in the background and the experimenter-defined locus. Though histones with a specific posttranslational modification may be detected in a reverse-ChIP experiment, identifying statistically significantly enriched histone modifications is much more difficult as a result of the high background signal due to the structural importance of histones in chromatin architecture. False negatives can result from proteins truly present at a locus of interest that are also expressed at high levels elsewhere. A more tractable experimental goal is the identification of uniquely bound molecular factors in the region, such as the presence of transcription factors that specifically regulate the region of interest and nowhere else. Contemporary reverse-ChIP experiments tend to focus on identifying molecular regulators uniquely present at the chosen locus and demonstrating the importance of those factors in regulating genomic expression.

Several demanding experimental difficulties not present in ChIP-Seq must be surmounted to develop a successful reverse-ChIP approach [30]. The issue of selective purification of factors associated with an arbitrary locus is the major experimental hurdle in the development of a standardized reverse-ChIP technique. ChIP-Seq relies on high-quality antibodies for purification. Even if there are no antibodies to a given factor, it can be engineered to incorporate an epitope tag, although doing so may disrupt natural binding properties [31]. In reverse-ChIP experiments, identifying unknown protein partners interacting with a specific genetic locus is the goal of the experiment, so antibodies cannot be employed for purification a priori. Researchers need to develop new biochemical tools for locus-specific purification.

Another issue in reverse-ChIP experiments is the high background. Unlike ChIP-Seq, which can include a PCR amplification step to increase signal, a reverse-ChIP technique must directly detect proteins present at the chosen locus. For some regulatory regions of interest (e.g., promoters or enhancers), there are only two copies of the locus per diploid cell (one per allele), while the remaining billions of base pairs in the genome and their associated protein complexes are sources of nonspecific background signal. A typical promoter is ~1 kilobase in length, meaning it comprises approximately 0.0001% of the 3 billion base pair human genome. However, a 1 kb promoter represents approximately 0.0083% of the 12 Mb yeast genome, meaning that there is more signal relative to the background for an individual, single-copy regulatory region in yeast. Accordingly, yeast genetics has been a testing ground for locus-specific chromatin isolation. Complicating matters further is the significant biochemical similarity that exists across all genomic regions: negatively charged DNA tightly winds around positively charged nucleosomes composed of histone proteins. Distinguishing proteins present at a specific locus from ubiquitously expressed chromatin structural proteins is a nontrivial challenge.

One strategy to overcome the difficulties associated with the high background is to target repetitive elements such as telomeres, which comprise ~0.01% of the human genome. Telomeres are repetitive regions that protect the ends of chromosomes during cell division and are critical to cancer biology. In healthy states, telomeres shorten as cells divide until they reach replicative senescence. Telomeres play a key role in cancer biology since cancer cells must activate mechanisms to maintain telomere length to allow for perpetual cell division [32,33,34,35,36,37,38,39,40]. Telomeres have served as a testing ground for developing strategies to isolate experimenter-defined chromatin regions for biochemical analysis.

Nonrepetitive genomic elements require high input volumes to generate enough signal for reliable detection above the background. Detecting the proteins associated with single-copy loci is the ultimate goal of reverse-ChIP experiments. The widespread utilization of alternative promoters is a common feature across many types of cancer [41,42,43,44]. In fact, pan-cancer transcriptome analysis has demonstrated that patient survival is correlated with the expression of the HER2 gene only from a specific promoter, not at the on/off gene-level expression [45]. A common mechanism for the expression of tumor-specific antigens is the activation of cryptic promoters within transposable elements near genes in cancer, highlighting the importance of specific promoters and their molecular context within the cancer phenotype [46]. A current frontier in personalized medicine for oncology is to target tumor-specific promoters [47]. The majority of disease risk variants identified in genome-wide association studies map to noncoding regions [48]; the pervasive utilization of alternative promoters and the promise of using those signatures as a therapeutic strategy is consistent with this observation. A reverse-ChIP strategy that enables biochemical pulldown of clinically-identified cancer promoters is a key technique in identifying the molecular mediators and potential therapeutic targets within the cancer regulome.

After optimizing the purification and reducing the nonspecific background, the detection and identification of purified factors still remain. Similarly to the development of ChIP-Seq, protein detection has evolved from a targeted, one-at-a-time approach to a proteome-wide quantitative technique. If a specific protein is hypothesized to interact with a purified locus, it can be detected by Western blotting. Detecting proteins via Western blot is a labor-intensive process requiring specific antibodies and a hypothesis that a factor binds there. Throughput is massively increased by using mass spectrometry-based proteomics for protein identification because it enables parallel detection and identification of thousands of proteins per sample in a discovery-based manner [49]. Modern mass spectrometers can detect and identify peptides present at the femtomole (10^−15^ moles) level. Therefore, at least several hundred million cells are required to detect a singly-bound protein present at a unique genetic locus (~10^8^ molecules). Fortunately, transcription factor binding sites are distributed periodically near gene promoters to increase the local concentration of regulatory factors, reducing the technical detection requirements beyond that demanding limit [50]. Nevertheless, experimental realities demand cell systems that can be easily grown to large scales and carefully designed controls to identify true interactors from nonspecific backgrounds. The phenotype of excessive cell growth embodied by cancer provides a technical advantage for locus-specific chromatin isolation since cells can be expanded to a large scale. This review first covers mass spectrometric techniques that enable quantitative proteomics. Subsequently, different approaches to selectively purify an arbitrarily defined genetic locus are described. The combination of biochemical isolation of a genomic locus with quantitative mass spectrometry-based proteomics represents a fully realized reverse-ChIP strategy.

## 2. Mass Spectrometry Techniques for Quantitative Proteomics

Mass spectrometry has become the tool of choice for proteomics experiments. Mass spectrometry-based proteomics is commonly used to study gene regulatory complexes [51,52]. The technique relies on the response of charged particles to a magnetic field in a vacuum to measure the mass-to-charge ratio (*m/z*) of the ion. Ionization is typically achieved via matrix-assisted laser desorption ionization (MALDI) or electrospray ionization (ESI), soft ionization techniques that prevent peptides from fragmenting before they can be analyzed. The resulting ions generate an MS spectrum characteristic of the analyte. Typically, the analyte is fragmented via collision with an inert gas, and the fragments are also analyzed by mass spectrometry. This technique, termed tandem mass spectrometry or MS/MS, provides unique fingerprints that are critical for accurately identifying the sequence of peptides of interest. A typical bottom-up mass spectrometry proteomics workflow (Figure 2a) begins with lysing cellular material (e.g., from cell culture, patient samples, etc.) [see [53] for an excellent introduction and primer]. Frequently, cell lysate is purified via SDS-PAGE to separate peptides by size to reduce sample complexity [54,55]. Subsequently, purified proteins are proteolytically digested to generate peptide fragments. Digested peptides are separated by physical properties and retention time using liquid chromatography. Eluted peptides are analyzed by a mass spectrometer to produce parent MS spectra before fragmentation to produce MS/MS spectra that facilitate sequence identification. A typical mass spectrometry experiment is run in data-dependent acquisition (DDA), in which individual ions are selected for fragmentation. An alternative strategy is data-independent acquisition (DIA), in which all ions present in a small *m/z* range are fragmented simultaneously to produce complex spectra [56]. Lastly, the data generated must be analyzed using database searching to identify proteins in the sample. Typically, a “decoy database” consisting of proteins with sequences reversed is also used as a control to detect the false positive rate. Spectra produced via DDA are simpler to analyze, but libraries are biased toward more abundant ions, whereas DIA results in spectra that are more difficult to analyze due to the sampling of every ion present.

Various strategies have been employed to modify the mass spectrometry procedure outlined above to enable quantitative proteomics. Different techniques employ interventions at a variety of stages, ranging from experimental perturbations that precede the described steps to alterations in sample preparation up to changes in data analysis. Broadly speaking, there are three major approaches to quantitative proteomics: (1) metabolic incorporation of heavy atoms by live cells to distinguish labeled and unlabeled samples, (2) labeling peptides by reacting them with isotopic chemical tags to enable comparison of different experimental conditions, and (3) label-free approaches that rely on data analysis for quantification. The logic underlying each technique and a few leading references are described below.

### 2.1. Metabolic Labeling

A commonly used strategy to isotopically label peptide samples for mass spectrometric analysis is to incorporate isotopes of heavy atoms metabolically as cells grow in culture (Figure 2b). Media supplemented with isotope-labeled arginine or lysine are commercially available and commonly used, in line with the sequence specificity of the frequently used protease trypsin (cleavage C-terminal to Lys or Arg). An advantage of this technique is that samples of different cell states can be combined before digestion and further processing of the proteins. This stands in contrast to chemical labeling strategies where different samples are processed in parallel until being combined just before the LC-MS step. Earlier combination of samples ensures any batch effects are evenly distributed across samples, which is not necessarily the case for chemical labeling strategies. Furthermore, there are fewer manipulations of samples after protein collection from cells. No reaction ever proceeds with 100% efficiency, so each additional sample handling step represents a potential for material loss. Metabolic labeling minimizes the number of steps from collection to analysis by incorporating isotopic labels as cells grow rather than after protein isolation.

#### 2.1.1. Stable Isotope Labeling by Amino Acids in Cell Culture (SILAC)

The prototypical example of metabolic labeling is stable isotope labeling by amino acids in cell culture (SILAC) [57]. Mammalian cells are grown in media lacking an essential amino acid. The media is supplemented with an isotopically labeled version of the missing amino acid so cells directly incorporate an isotopic label during growth and protein synthesis. In the initial report on the SILAC technique, tri-deuterated leucine was wholly incorporated into all cells within five doublings. Combining heavy atom-labeled cells with cells grown in standard media enables a direct ratiometric comparison of protein abundance between two conditions. Signals of proteins in both conditions are doubled in MS mode with a separation between peaks determined by the *m/z* difference between the labeled and unlabeled peptides. The authors demonstrate the utility of SILAC by measuring changes in protein abundance during muscle cell differentiation. Analysis of SILAC data can be efficiently quantified using the MaxQuant suite of algorithms [58].

#### 2.1.2. Triple-State SILAC

The SILAC technique can be extended to compare three states by utilizing differentially labeled amino acids. The labeled amino acid can include one or two isotopic labels, creating light (unlabeled), medium (singly labeled), and heavy (doubly labeled) media [59]. The three states can be mixed to give ratios of L/M, H/M, and H/L to directly quantify across conditions. Using the same reference sample across experimental replicates enables extension beyond the three states. In the initial triple SILAC report, dynamics of protein signaling in response to EGF stimulation were measured across five time points with unstimulated cells serving as the common control. Triple SILAC can also determine protein interactions in response to a stimulus by having unstimulated wild-type cells in one condition, unstimulated cells with an affinity tag included on the protein of interest in a second condition, and stimulated cells with an affinity tag in the final condition [60]. This approach was used to study interactions in the Wnt signaling pathway. The ratio of stimulated to unstimulated cells gives the response to the stimulus while the ratios of the affinity-tagged samples to wild-type cells separate specific binders from background nonspecific binding.

#### 2.1.3. Isotopic Differentiation of Interactions as Random or Targeted (I-DIRT)

The SILAC protocol has been adapted to measure not just changes in protein expression between cell states but to distinguish between specific and nonspecific interactions of proteins. The technique is termed isotopic differentiation of interactions as random or targeted (I-DIRT) [61]. A cell line is modified to incorporate a purification handle on a protein of interest. These cells are grown on light isotopic media, while wild-type cells are grown on heavy media. Cells are mixed in a 1:1 ratio before cryogenic lysis and subsequent affinity purification utilizing the introduced tag. When the peak of a peptide is doubled in the mass spectrum, it indicates the protein was present in both heavy and light samples and represents a nonspecific contaminant. Only peptides identified solely with the light label are true, stable interaction partners. The addition of a formaldehyde cross-linking step allows for the detection of true transient interactions for peaks that are doubled but not present in a 1:1 ratio [62]. I-DIRT relies on combining cell states before lysis, a unique advantage of SILAC and metabolic labeling. The approach is broadly applicable to purification methods beyond the incorporation of an affinity tag (vide infra).

### 2.2. Chemical Labeling 

A variety of approaches have been employed to enable quantitative mass spectrometry-based proteomics leveraging chemical modification of protein samples (Figure 2c). These techniques rely on the modification of protein/peptide samples after isolation from cells. During sample preparation, a chemical handle in the form of an isotopic label is incorporated into separate samples, which are then combined for mass spectrometric analysis. Samples labeled with heavy atom tags are distinguishable from samples labeled with light tags due to the difference in *m/z* in MS mode or MS/MS mode, depending on the specific tag being used. Because the only chemical difference in isotopically labeled tags is the number of neutrons contributing to the atomic mass, differentially labeled samples show the same efficiency of ionization, a key parameter for accurate quantitative comparisons. This technique is compatible with samples that cannot be grown in cell culture. Chemical labeling techniques allow for sample multiplexing. Different chemical labels have their own residue-specific reactivity profiles, which have been identified with the computational tool pChem [63].

#### 2.2.1. Isotope Coded Affinity Tags (ICAT)

The first report of chemical labeling for mass spectrometry-based proteomics was the use of isotope-coded affinity tags (ICAT) [64]. The ICAT reagent consists of three major elements: an iodoacetamide group to specifically react with cysteine residues, a linker labeled with hydrogen (light) or deuterium (heavy) atoms, and a biotin purification handle. Peptides from one cell state are labeled with the light ICAT reagent while peptides from a separate cell state are labeled with the heavy reagent before both samples are mixed together. The biotin handle enables selective purification of labeled peptides, an important step to reduce sample complexity before mass spectrometric analysis. Subsequently, the mixed samples are analyzed by LC-MS. The eight deuterium atoms present in the heavy linker cause a shift in *m/z* compared with the light label in MS mode. A ratiometric comparison between the two samples provides relative quantification between two cell states. In the initial report of ICAT reagents, the authors measure differences in protein expression in yeast using either galactose or ethanol as the carbon source. A major issue in utilizing deuterium for isotopic labeling is the partial resolution of isotopic pairs during the LC step. ICAT reagents are therefore modified to incorporate ^13^C as the heavy atom label rather than ^2^H, preventing isotopic resolution during purification [65].

#### 2.2.2. ^18^O Metabolic Labeling and Dimethyl Labeling

Although ICAT reagents have provided a step forward for quantitative proteomics, detection is limited to peptides containing cysteine. Alternate chemical reactivity following the same strategy of a ratiometric comparison of isotopically labeled samples was therefore developed to expand the fraction of the proteome accessible to quantitative mass spectrometry. One approach is to label samples with ^18^O via enzymatic digestion of proteins in the presence of heavy water [66]. Digested samples incorporate two ^18^O atoms. Combining these heavy atom-labeled samples with peptides enzymatically digested in isotopically abundant water allows for a ratiometric comparison between samples due to the 4 Da mass difference. This technique has been used to compare the proteome of two serotypes of adenovirus [67]. An alternative approach to expanding chemical tags beyond cysteine-containing peptides is to utilize dimethyl labeling [68]. This technique uses isotopically labeled formaldehyde to perform a reductive amination on all free amines in the peptide sample. In addition to lysine side chains, all peptide fragments produced by enzymatic digestion contain free amines at the *N*-terminal end. Thus, the dimethyl labeling strategy can label any protein regardless of its sequence composition. The speed of labeling and relative cheapness of labeled formaldehyde compared with other isotopic sources are advantages of this technique. The initial report from the authors also demonstrates the co-elution of heavy- and light-labeled peptides.

#### 2.2.3. Tandem Mass Tags (TMTs)

A shortcoming of the techniques described above is the increased complexity of MS spectra upon chemical labeling. When a peptide is present in both heavy- and light-labeled samples, its signal is duplicated in the parent MS spectrum due to the difference in mass between the heavy and light labels. This issue was addressed through the development of isobaric tandem mass tags (TMTs) [69]. Similar to ICAT reagents and dimethyl labeling, TMT experiments label separate cell states in isolation before pooling them together for mass spectrometric analysis. The difference with TMTs is that all tags have the same mass in parent MS mode (i.e., they are isobaric). The isotope distinguishing the labels varies its position, leading to different fragmentation patterns that can be detected in MS/MS mode to allow for ratiometric quantification between samples. The tag consists of three major elements: (1) an amine-reactive moiety to label *N*-terminal peptide ends and lysine residues, (2) a reporter ion with a characteristic mass measured in MS/MS mode, and (3) a mass normalization linker to produce identical parent MS spectra for the same labeled peptides from different samples. Heavy atoms, such as ^13^C, ^15^N, or ^18^O, are incorporated at different positions in the reporter ion and mass normalization linker, so each reporter ion produces the same signal in MS mode but a unique signal in MS/MS mode. Because the chemical tags have the same mass in parent MS mode, signals from the same peptide in different samples are not doubled and spectra are simpler to analyze. The mass of the TMT reagent is then chosen for fragmentation in MS/MS mode, and intensities of the reporter ions are a direct measure of abundance, allowing for ratiometric quantification.

#### 2.2.4. Isotope Tagging for Relative and Absolute Quantification (iTRAQ)

The general strategy of utilizing isobaric tags with an amine-reactive linker, an isotopically labeled reporter ion, and a mass normalization linker can be adapted using a variety of molecular structures. An additional advantage of the strategy of using isobaric mass tags is the enhanced capacity for sample multiplexing. Sample 4-plexing was demonstrated with the report of isotopic tagging for relative and absolute quantification (iTRAQ) reagents to compare protein expression in wild-type yeast with two mutant varieties [70]. The chemical structure of the isobaric tag enabled the incorporation of isotopic labels in different positions on the reporter ion and mass normalization group to increase the number of samples that could be studied simultaneously. The inclusion of a synthetic peptide sample of known concentration during analysis enabled not just relative quantification of protein abundance between samples but absolute measurements as well. Modern mass spectrometers can even detect the mass difference between isobaric tags labeled with ^13^C vs. ^15^N to increase multiplexing capability [71]. Though the presence of ^13^C or ^15^N in the reporter ion both represent the addition of a single neutron, there is a mass difference of 6.32 mDa due to the difference in nuclear binding energy between the two atoms that can be reliably measured. Recent developments have produced TMT reagents capable of 16-plexing and 18-plexing [72,73,74]. A comparison of different isobaric labeling techniques and considerations in experimental design has been reviewed elsewhere [75].

#### 2.2.5. Ratio Compression in Isobaric Tags Solved with MS3 and Synchronous Precursor Selection

The greatly enhanced capacity for multiplexing enabled by isobaric mass tags is a major advantage of these approaches. Unfortunately, empirical observations have demonstrated that measured ion ratios are compressed relative to known quantities, resulting in an underestimation of peptide abundance changes between samples [76,77]. The underlying cause of ratio compression is multiple ion interference. During the acquisition and selection of ions to fragment for MS2, contaminating ions with similar mass are co-isolated and fragmented as well, skewing the observed ratios from the isobaric reporter tags away from the true value. Although the parent ions selected for MS2 quantification have nearly identical masses to the contaminating ions, the differing underlying molecular structures give rise to different fragmentation patterns. Accordingly, an additional fragmentation step (i.e., MS3) can increase specificity and eliminate ratio compression [78].

The increased specificity afforded by triple-stage mass spectrometry (MS3) comes at a cost of decreased sensitivity. The number of proteins quantified in an MS3 experiment was 12% lower than in a standard MS2 experiment, reflecting both the lowered intrinsic signal (only a fraction of MS2 ions were fragmented to produce MS3 spectra) and increased acquisition time to generate MS3 spectra. The former limitation can be addressed through synchronous precursor selection (SPS) [79]. Instead of selecting a single ion from the MS2 scan to fragment to generate the MS3 spectrum, multiple fragment ions are co-isolated to increase the number of MS1 precursor ions that convert into MS3 TMT reporter ions. The isolation waveform for the acquisition of MS3 spectra is designed in a way to co-isolate multiple fragments in the MS2 spectrum that are generated by the same parent ion from MS1 while excluding contaminating ions. The synchronous precursor selection method allows for ~40% of MS2 ions to be collected in MS3, which results in ~25% of the initial MS1 precursor population converting to the MS3 precursor population (compared with ~5% in a traditional MS3 approach). Real-time database searching is a strategy that has been developed to ameliorate the decreased sensitivity due to increased acquisition times resulting from the longer duty cycle to generate MS3 scans [80,81]. Spectra produced in MS2 are simultaneously compared against a database. Only ions that generate peptide spectral matches are selected for fragmentation in MS3 vis synchronous precursor selection; ions that cannot be confidently assigned to a peptide are not selected for SPS-MS3. This reduction in the collection of spurious SPS-MS3 spectra can cut instrument time by half compared with traditional SPS-MS3 without online real-time database searching.

### 2.3. Label-Free

An alternative approach to quantitative mass spectrometry proteomics is to forgo the use of labels entirely (Figure 2d). This holds the advantage of avoiding the use of expensive isotopically labeled reagents and is compatible with primary samples that cannot be grown in cell culture. However, data analysis is computationally demanding.

#### 2.3.1. Peak Area Measurements

One label-free technique quantifies proteins based on peptide peak areas from the chromatogram produced during the LC step. Since the retention time of peptides is based on their physical properties, it is reproducible across LC runs. The size of the chromatogram peak is a measure of the abundance of the identified peptides. Integration of the peak produced in the chromatogram that gives rise to characteristic MS1 spectra (and the associated MS2 fragmentation patterns) directly quantifies identified peptides. In the initial report, peak area intensity measurements could detect proteins at concentrations as low as 10 fmol across a 10^4^ dynamic range [82]. A key consideration in accurate quantification is the correct alignment of peaks across LC runs based on retention time, *m/z*, and spectra in MS and MS/MS modes. The development of algorithms such as MaxLFQ enhances the accurate identification and quantification of peaks [83].

#### 2.3.2. Spectral Counting

An alternative label-free approach relies on the sampling of peptides to produce MS/MS spectra during the LC-MS run. The number of ions present during LC runs of complex mixtures exceeds the ability of mass spectrometers to produce MS/MS spectra for all ions before they elute. Peptides present at higher concentrations are sampled more frequently, so spectral counting is a reliable method for protein quantification [84]. Spectral counts have been used to quantitatively identify differential protein expression of mouse cochlear sensory epithelial cells as hearing developed [85]. A related approach is to normalize the number of peptides observed during an LC-MS run to the number of peptides that could be observed from the parent protein. This ratio (protein abundance index [PAI]) is proportional to the logarithm of protein concentration and can provide accurate absolute quantification of proteins with an exponential modification (emPAI) [86]. A comparison of label-free methods against each other has been reviewed elsewhere [87,88].

### 2.4. Comparison of Quantitative Mass Spectrometry Techniques

Metabolic labeling and isobaric tags both provide approaches to compare protein levels quantitatively across samples. Isobaric tags provide an experimental paradigm that enables much greater sample multiplexing in comparison with SILAC. However, the issue of co-isolation of contaminating ions can cause ratio compression that distorts measured abundances from their true values for isobaric tagging-based experiments. This problem can be ameliorated through the use of an additional fragmentation scan (MS3) with the tradeoff of increased instrument acquisition time. Though SILAC does not have the same multiplexing capabilities as isobaric tagging, this technique holds the advantage that isotopic labels are incorporated at the cell growth stage, allowing for samples to be mixed before performing any biochemistry. The ability to combine samples and process them simultaneously (rather than in parallel as for isobaric tagging) holds unique technical advantages that have been leveraged by researchers pursuing locus-specific chromatin isolation strategies.

Label-free methods have been directly compared with metabolic labeling and chemical labeling to determine the best technique for quantitative proteomics [89]. Spectral counting provided the deepest proteome coverage for identification but performed worse in quantitative measurements, especially for low-abundance proteins. Both metabolic labeling and chemical labeling provided accurate, reproducible quantification, with isobaric chemical labeling providing higher precision and reproducibility. For isobaric chemical labeling, TMT and iTRAQ reagents were comparable. A separate comparison of iTRAQ 8-plexing with the label-free methods of spectral counting, peak area, and emPAI identified label-free peak area measurements as superior for small abundance changes (as low as 1.1-fold difference) [90]. Spectral counting performs poorly with low-abundance proteins and small fold changes but works well in other conditions. iTRAQ reagents were outperformed by label-free methods for low-abundance proteins but showed good quantification in other regards. Label-free methods, metabolic labeling, and isobaric chemical tags all provide an experimental paradigm to accurately identify and quantify protein levels after isolation from cells. The frequently used MaxQuant software, which has incorporated the MaxLFQ workflow, has been updated since its release to allow for a computational analysis of quantitative mass spectrometry data utilizing any of the above-described workflows [91]. MaxQuant has also been adapted to enable the analysis of DIA mass spectrometry data [92]. Quantitative mass spectrometry techniques are a necessity for building an understanding of the complete stoichiometric composition of a locus-specific proteome. However, even nonquantitative mass spectrometry can be useful to identify novel genome regulators at a locus of interest with the appropriate controls. A protein identified in experimental samples but not in control samples is a good candidate for a locus-specific regulator to be tested with follow-up functional experiments.

## 3. Reverse-ChIP Approaches for Locus-Specific Chromatin Isolation

A key aspect of any reverse-ChIP technique is the ability to biochemically isolate the genetic locus of interest from background chromatin. Distinguishing low-abundance proteins enriched at a specific locus from the background necessitates the development of stringent purification techniques in addition to advances in mass spectrometry. Three general approaches have been employed to selectively purify a specific genomic locus to detect associated proteins: (1) nucleic acid-based approaches, (2) protein-based approaches, and (3) CRISPR-based approaches. Different experimental approaches are summarized in Table 1. 

### 3.1. Nucleic Acid-Based Approaches

In developing purification strategies for a specific genomic locus, an experimental handle exists intrinsic to the system: the sequence of the region itself. A variety of approaches have been employed leveraging synthetic nucleic acid probes as handles for the enrichment of interaction partners at a specific locus (Figure 3). These approaches hold the advantage of obviating the need for laborious genetic engineering.

#### 3.1.1. Synthetic Oligo Baits

One such approach is to add biotin to the end of an oligonucleotide synthesized to match the sequence of a region of interest (Figure 3a). Proteins interacting with the sequence can be purified from cell lysate with streptavidin beads. Mutations induced into the sequence of interest distinguish nonspecific background interactors from true partners. There is no cross-linking step in this approach: the synthetic oligo baits are used to capture free proteins in cell lysate that interact with the chosen sequence. A critical experimental development is the ability to produce analytical scale sample volumes from the high initial number of input cells [93] coupled with a mass spectrometric readout [94]. Before this advance, protein complexes would dissociate in the non-equilibrium high-volume conditions, resulting in false negatives for indirect interaction partners (that is, proteins specifically recruited to a locus that does not directly interact with the DNA sequence). Coupling this strategy with quantitative mass spectrometry enables the discovery of new biology. Using an ICAT-based approach, researchers were able to identify the protein binding factor interacting with the muscle creatine kinase enhancer [95]. Quantitative mass spectrometry was critical for success in this technique: of the nearly 900 proteins purified via interaction with the nucleic acid bait, only three showed >2-fold difference in abundance with wild-type vs. mutated bait. New proteins recognizing methylated DNA were identified by using a methylated vs. nonmethylated oligonucleotide bait in a SILAC-based approach [96]. Label-free quantitative proteomics identified allele-specific regulation of the *PPARG* locus using biotinylated probes representing a risk and non-risk allele [97].

Quantitative mass spectrometry for analysis of proteins purified from cell lysate via synthetic oligo baits can serve as a powerful tool to understand cellular phenotype and epigenetic regulation. Researchers synthesized a defined oligo sequence containing unmodified cytosine, methylated cytosine, or various oxidized derivatives of methyl cytosine [98]. Using a mixture of two- and three-state SILAC, the scientists identified protein readers of each derivative of cytosine in mouse embryonic stem cells. Most readers showed a strong preference for specific chemical modifications. DNA repair pathway enzymes are associated with 5-formylcytosine, while there are unique readers associated with both less- and more-oxidized forms of methyl cytosine (i.e., 5-hydroxymethylctosine and 5-carboxycytosine), indicating that different chemical modifications of DNA bases are bona fide epigenetic marks. A label-free approach utilizing MaxQuant identified readers of chemically modified cytosine bases in mouse neural progenitor cells and adult mouse brain cells. Unique readers for each of the chemically modified cytosine bases were found in all cell types studied. There was limited overlap between the different cell states, indicating a dynamic change in readers of epigenetic marks as cells progressed through differentiation. Identifying the readers and writers of cancer-specific epigenetic marks is a promising avenue for identifying therapeutic targets for clinical translation [99].

#### 3.1.2. Proteomics of Isolated Chromatin (PICh)

The development of a reverse-ChIP protocol marked a major advancement with the development of the proteomics of isolated chromatin (PICh) technique [100]. PICh represents a conceptual shift in the purification of factors associated with a specific genomic locus: rather than providing a nucleic acid bait to allow protein factors to interact with the sequence of interest, PICh directly isolates chromatin and associated factors (Figure 3b). Cells are cross-linked with formaldehyde to covalently link protein complexes to chromatin. Purification is achieved via a nucleic acid probe attached to desthiobition with a linker. Desthiobiotin is an analog of biotin that binds tightly to streptavidin and other biotin-binding proteins but is able to be competitively displaced by biotin under mild conditions [101]. Approximately half of the oligonucleotide probe is composed of locked nucleic acids, allowing for direct hybridization of the probe to the targeted chromatin region due to the increased melting temperature of the probe-DNA hybrid. The cross-linking step and increased probe-DNA hybrid stability enable direct isolation of the targeted chromatin via the biotin handle. Since the technique utilizes nucleic acid probes to obtain proteomic information, it is named PICh to highlight the contrast with ChIP, which uses protein probes to obtain sequence information. 

As a proof of concept, the original PICh report targeted telomeres in human cells due to their high abundance (~100 telomeres/cell). Both known and unknown factors associated with telomeres were enriched using telomere-specific probes compared with scrambled probes. The authors compared proteins found at telomeres maintained by the alternative lengthening of telomeres (ALT) pathway compared with those maintained by telomerase. The ALT pathway is one strategy cancer cells use to protect their telomeres [102]. Previously identified telomere-associated proteins were found, validating the technique. Proteins previously unidentified with telomeres were also identified by mass spectrometry and validated via orthogonal techniques such as fluorescence colocalization. Orphan nuclear receptors that previously had no assigned function were identified as telomere-associated proteins in cells utilizing the ALT pathway. 

A subsequent report targeted telomere-associated sequence repeats in *Drosophila melanogaster* [103]. These sequence repeats have approximately 30-fold fewer hybridization positions compared with human telomeres and, therefore, represent a step forward in the power of the PICh technique. An important lesson learned in this report is that purification from the background is more dependent on having a cluster of closely-spaced hybridization sites than on an abundance of sites alone. A study from a separate research group using PICh to study centromeres in barley identified seven H2A variants associated with the centromere-specific PICh probe that were not present in the scrambled probe [104]. The robustness of the strategy of hybridizing nucleic acids directly to chromatin for purification was demonstrated by another research group, extending the approach to a single-copy locus with the identification of hundreds of potential regulators of the *AtCAT3* promoter in *Arabidopsis thaliana* by mass spectrometry in a technique the authors call RChIP [105]. Five randomly selected transcription factors identified from RChIP were verified to interact with the promoter through conventional ChIP-qPCR and electromobility shift assay (EMSA) analysis.

#### 3.1.3. Modified PICh (qPICh and ePICh) 

Since the initial disclosure of the technique, PICh has been further developed in subsequent reports. Combining PICh with SILAC led to quantitative PICh (qPICh) [106]. Authors used qPICh to study pericentromeric regions in mouse embryonic stem cells. The quantitative comparison across conditions enabled by the SILAC technique led to the discovery that BEND3 is a key factor in the transition from constitutive to facultative heterochromatin, shedding light on a previously obscure mechanism. A separate qPICh study revealed novel insights into the ALT pathway [107]. In contrast to the previously prevailing view that the formation of heterochromatin inhibits ALT, researchers demonstrated that the action of the histone methyltransferase SETDB1 to seed heterochromatin formation is a crucial factor for cells to transition to the ALT pathway. Counterintuitively, the seeding of heterochromatin was associated with an increase in transcription. This study enabled by qPICh points toward the possibility of SETDB1 inhibitors as a novel therapeutic approach to cancer treatment.

During the optimization of PICh protocols, researchers found that purification was more successful when the locked nucleic acid probes targeted the ends of a stretch of DNA compared with internal targeting. This led to the creation of end-targeting proteomics of isolated chromatin (ePICh), in which restriction enzymes are used to create defined chromatin fragments rather than the random ends produced by sonication [108]. Using this and other modifications to the conventional PICh protocol, the authors purified the chromatin associated with a ribosomal RNA promoter and identified a novel factor that targets RNA polymerase I to the promoter. The authors indicate that the degree of enrichment obtained for this ~200 copy, 1-kb locus in the mammalian genome approximately corresponds to a 1-kb single-copy locus in the yeast genome.

#### 3.1.4. Hybridization Capture of Chromatin-Associated Proteins for Proteomics (HyCCAPP) 

Conceptually similar to the PICh approach is a technique termed hybridization capture of chromatin-associated proteins for proteomics (HyCCAPP) developed by another research group [109]. Like PICh, this technique utilizes nucleic acid probes covalently linked to desthiobiotin to directly purify a sequence-defined chromatin region and identify associated proteins using mass spectrometry. The initial reports of HyCCAPP were performed in yeast, including isolation of the single-copy *GAL1-10* promoter. Researchers identified known and novel protein partners interacting with each of the loci under study. A subsequent HyCCAPP study using spectral counting for quantitative mass spectrometry identified differential protein occupancy at the *ENO2* promoter in yeast grown on glucose vs. galactose [110]. An extension of HyCCAPP to mammalian cells studying centromeric alpha satellite DNA in human K562 cells showed good agreement with PICh results from mouse embryonic stem cells [111]. This study utilized label-free ion chromatogram peak area measurements. Considerations for adaptation of HyCCAPP in mammalian cells vs. the yeast system it was originally optimized in have been reviewed elsewhere [112]. A sequential probe release strategy enabled a multiplexed HyCCAPP approach that allowed for the capture of multiple single-copy loci in a single experiment, expanding the experimental capabilities of the system [113]. In these HyCCAPP reports, both known and novel proteins were significantly enriched at the target locus and confirmed via orthogonal experimental methods.

#### 3.1.5. Global ExoNuclease-Based Enrichment of Chromatin-Associated Proteins for Proteomics (GENECAPP)

An alternative nucleic acid-based strategy has been developed termed global exonuclease-based enrichment of chromatin-associated proteins for proteomics (GENECAPP) (Figure 3c) [114]. This technique was initially developed using an in vitro system studying the IGFBP1 promoter and FoxO1, a known binder to the region. After formaldehyde cross-linking and DNA fragmentation, an exonuclease was used to generate a single-stranded DNA region. The desired sequence and associated proteins could be pulled down using a bead-attached sequence-specific probe. The in vitro system allowed the authors to optimize several experimental parameters, including formaldehyde concentration, cross-linking time, and quenching agent. A key finding was that quenching formaldehyde with Tris was more effective than with the traditional glycine quench. A separate group adapted GENECAPP to an in vivo system to study the *dsz* promoter in desulfurizing bacteria [115]. Proteins identified by mass spectrometry were demonstrated to directly bind to the sequence representing the targeted region through EMSA.

### 3.2. Protein-Based Approaches

Though nucleic acids provide an experimental handle to specifically target an arbitrary genomic locus, purification efficiency is low. For ePICh, a single nucleic acid probe is not sufficient to purify a targeted chromatin segment, requiring the use of several purification handles. A typical PICh experiment uses as many as 50 15-cm^2^ plates per condition to address this problem. By contrast, protein-based purification can have higher sensitivity. Rapid immunoprecipitation of endogenous proteins (RIME), an antibody-based ChIP strategy, can detect novel proteins involved in chromatin complexes with as few as 10^6^ cells [116]. The widespread adoption of ChIP-Seq has led to the development of standardized workflows and quality controls for antibody-based purification [117]. Therefore, researchers have developed techniques that utilize proteins as a purification tag for a genomic locus (Figure 4). This technique often follows a two-step process: first, the region of interest is genetically engineered to incorporate an experimenter-defined exogenous binding element (EBE). Subsequently, the engineered cell line is further engineered to include an exogenous recognition element that binds to the experimenter-defined EBE already introduced.

#### 3.2.1. Recombinase Excision

An early approach to purifying a specific genomic locus with protein-based methods relied on genetic engineering and recombination. Experimenters engineered recombinase sites to flank the region of interest, enabling the segment to be excised into a small chromosomal circle that retained native interactions (Figure 4a) [118]. Subsequent centrifugation separated the region of interest from bulk chromatin, enabling functional studies. A 2.5 kb region of the transcriptionally repressed *HMR* locus in yeast could be isolated via this technique. The specificity of purification was improved with a subsequent modification to the procedure by another group. In addition to the recombinase sites, the authors inserted the binding sequence of the bacterial LexA protein into the *PHO5* gene in yeast [119]. Purification with a LexA antibody after centrifugation further specifically enriched the target locus. Both a 750 bp promoter region and the entire 2200 bp gene region were successfully targeted for purification. In the initial report, the authors showed differential chromatin accessibility to restriction enzymes based on whether the gene was transcriptionally active or silent. Subsequent improvements to the purification technique enhanced sensitivity to the point that quantitative proteomics using iTRAQ reagents became possible [120]. The authors identified previously known protein interactors and orthogonally confirmed the interaction of unknown proteins with the specific locus under study.

#### 3.2.2. Insertional Chromatin Immunoprecipitation (iChIP)

An advancement in protein-based purification strategies came with the development of insertional chromatin immunoprecipitation (iChIP) [121]. Building off developments in recombinase excision, this technique relies on the introduction of a known exogenous sequence handle into the genetic region of interest (Figure 4b). Expression of the protein recognizing the introduced sequence modified to include an affinity tag allows for purification of the genomic region. Neither the recognition sequence nor the binding protein exists in wild-type cells, providing a negative control to identify nonspecific contaminants. This approach obviates the need for recombinase excision and centrifugation, allowing for the isolation of the region of interest directly from bulk chromatin. In the first report of the iChIP strategy, researchers introduced eight repeats of the LexA-binding sequence to purify the IRF-1 promoter from BLG cells using a FLAG-tagged LexA-binding protein. PCR amplification coupled with appropriate negative controls demonstrated successful purification of the genomic region. The authors were also able to identify proteins purified from the region with 10^6^ cells by using a sequential immunoprecipitation strategy. After purifying the tagged region with an anti-FLAG antibody, the chromatin was purified with an antibody against Stat1, a known protein binding to the locus. Stat1 was detected by Western blotting, demonstrating the feasibility of using iChIP as a reverse-ChIP strategy. A key step in any iChIP experiment is performing control experiments to demonstrate that the introduction of the DNA recognition sequence does not disrupt the regulatory function of the region under study.

The iChIP strategy is compatible with any DNA sequence and protein recognition partner that are not present in wild-type cells, which has led to its adoption by a number of research groups. An iChIP strategy utilizing the bacterial LacO sequence and a tagged LacI-binding protein found distinct functions for two regulatory proteins in yeast previously thought to be redundant [122]. A LacO-based iChIP study in *Drosophila* embryos correlated chromatin conformation with gene expression in a cell type-specific manner [123]. Though these studies utilized sequencing-based readouts, they demonstrated the ability to isolate sequence-defined genetic regions. The *Drosophila* study was also able to identify proteins with a sequential immunoprecipitation strategy as described above. Optimization of the tagged LexA protein [124] and use of recombinant protein [125] increased the efficiency of isolation and allowed for purification of single-copy loci without needing to express the protein in cells, extending the capabilities of the iChIP by removing one of the genetic engineering steps. The combination of iChIP with mass spectrometry allowed for the identification of new proteins involved in the regulation of a region of interest. A LexA-based iChIP study of the β-globin HS4 insulator led to the identification of new proteins binding to the region using 10^7^ cells engineered to contain 24 copies of the locus [126]. RNA species interacting with the region were also identified after purification via sequencing. A comparison of iChIP with other techniques to study genomic structure has been reviewed elsewhere [127].

The combination of iChIP with quantitative mass spectrometry allowed for the comparison of gene regulation across cell states. An iChIP-SILAC study of the single-copy *Pax5* locus in chicken B cells led to the identification of two B cell-specific proteins regulating the expression of Pax5 [128]. B cells engineered with the LexA-binding element were grown in heavy isotopic media, whereas transdifferentiated cells were grown in light isotopic media to identify cell-state specific proteins at the *Pax5* gene. Mass spectrometry results provided the seeds for functional studies to identify Thy28 and MYH9 as B cell-specific proteins integral to controlling Pax5 expression levels.

#### 3.2.3. Chromatin Affinity Purification Mass Spectrometry (ChAP-MS)

In a separate study, iChIP was combined with i-DIRT (vide supra) to study the regulation of the single-copy *GAL1* locus in yeast cells under transcriptionally active and silent states [129]. A LexA-Protein A fusion protein was constitutively expressed in yeast cells. Cells with a LexA-binding sequence engineered upstream of the *GAL1* start codon were grown in light media while cells without the binding element were grown in heavy media. The authors refer to this technique as chromatin affinity purification mass spectrometry (ChAP-MS). ChAP-MS identified novel histone modifications, including several for which there are no validated antibodies. The unbiased purification strategy coupled with mass spectrometric quantification allows for the identification of proteins that could not be studied using conventional ChIP-Seq.

#### 3.2.4. Targeted Chromatin Purification (TChP)

An alternative strategy termed targeted chromatin purification (TChP) identified transcription factors involved in the repression of γ-globin [130]. This technique is conceptually similar to iChIP and ChAP-MS with an additional layer of chemical control (Figure 4c). The inserted EBE for TChP is the tet-responsive element (TRE), and its binding partner is the TetOn protein, which only interacts with the TRE in the presence of doxycycline. In the initial report, the TetOn protein was modified to incorporate a site for in vivo biotinylation by the BirA biotin ligase, giving a handle to purify with streptavidin magnetic beads. An advantage of TChP is that the engineered binding site is present in both control and experimental samples. The presence or absence of doxycycline determines whether or not the purification handle will bind to identify locus-specific proteins or nonspecific background binders, providing a clean chemical control. Proteins proposed to interact with a locus by chromatin purification coupled with mass spectrometry must be confirmed through orthogonal experimental techniques. Knockdown of transcription factors identified by TChP directly led to increased expression of γ-globin, demonstrating functional relevance.

An adaptation of TChP performed in another laboratory with an additional layer of chemical control was utilized to uncover regulators of the core promoter of the hepatitis B Virus (HBV) [131]. The TetR-binding protein was fused to FRB, which can form a ternary complex with FKBP12 in the presence of rapamycin [132,133]. Attached to FKBP12 was TurboID, an engineered enzyme for proximity labeling, a technique in which biotin molecules are directly bonded to nearby proteins (vide infra) [134]. Mass spectrometry and functional studies identified STAU1 as a key regulator of HBV. Interestingly, STAU1 did not show interaction with the region via EMSA. Further experiments demonstrated that STAU1 was indirectly bound to the region, recruited by TARDBP and recruited the SAGA transcriptional activator complex to the region in turn. The precise chemical control afforded by doxycycline to induce binding to the TRE and rapamycin-induced binding of the proximity labeling enzyme gives layers of control over the experiment, reducing the false positivity rate.

#### 3.2.5. Transcription Activator-Like (TAL) Protein Targeting

A drawback of iChIP and related strategies is the need to engineer an exogenous DNA sequence in the region of interest followed by the recognition element that binds to the region for purification. The introduction of the EBE coupled with the necessary controls to demonstrate that the regulatory function of the region under study has not been disrupted is a laborious task. To circumvent this problem, researchers turned to transcription activator-like (TAL) proteins [135]. TALs are modular proteins that can be designed to recognize any sequence of interest, precluding the need to engineer the target locus. The two-step genetic engineering of cell lines turns into a single step (Figure 4d). Locus-specific chromatin isolation strategies utilizing TALs have been adopted by multiple independent research groups.

##### Engineered DNA-Binding Molecule-Mediated Chromatin Immunoprecipitation (enChIP) and TAL-ChAP-MS

The promise of TALs to simplify experimental design was recognized by laboratories already working on chromatin isolation techniques. A TAL protein was designed to recognize telomere repeats. This technique was called engineered DNA-binding molecule-mediated chromatin immunoprecipitation (enChIP) as a callback to the iChIP approach previously developed in the lab [136]. Both known and novel telomere-interacting proteins were identified by MS using 10^7^ cells. Telomere-associated RNAs were also identified with a sequencing-based readout. Novel proteins were demonstrated to co-localize with telomeres via live cell imaging, demonstrating the utility of enChIP to discover new biology. The enChIP technique was brought to a single-copy locus by a separate research group to study regulators of the proximal enhancer of OCT4, a key regulator of pluripotency in human embryonic stem cells [137]. A total of 10^11^ cells were used to pull down the enhancer with a 3x-FLAG TAL, from which 150 nuclear proteins were identified by mass spectrometry. The identification of known regulators of pluripotency gave confidence in the technique. A subsequent siRNA knockdown of transcription factors identified and confirmed to bind to the region through ChIP-qPCR led to significant changes in OCT4 mRNA expression. The protein ZNF207 emerged as a previously unknown regulator of self-renewal and pluripotency, as well as a key factor in differentiation toward the ectoderm fate. A comparison of iChIP vs. enChIP has been reviewed elsewhere [138]. In a separate report, the *GAL1* locus was studied in yeast by TAL-ChAP-MS to enable a direct comparison with ChAP-MS using a PrA-tagged engineered TAL protein [139]. Enrichment of the locus under transcriptionally active conditions was comparable between the two techniques. Isolation of the region under transcriptionally silent conditions was not observed with TAL-ChAP-MS, suggesting the possibility of differentially accessible DNA under different transcriptional states. TAL-ChAP-MS used label-free techniques for quantitative mass spectrometry, demonstrating the feasibility of reverse-ChIP protocols that do not rely on expensive isotopic labeling.

##### TAL-Mediated Isolation of Nuclear Chromatin (TINC)

The modularity of TALs has given an opportunity for protein-based reverse-ChIP strategies to be adopted more widely. Researchers have turned to studying regulation at the promoter of *Nanog*, a key regulator of pluripotency, via TAL-mediated isolation of nuclear chromatin (TINC) [140]. In this experimental design, mouse embryonic stem cells were engineered to stably express two epitope-tagged TALs in separate cell lines. Each TAL targeted a unique sequence in the *Nanog* promoter just upstream of the binding sites of Oct4 and Sox2. Only proteins enriched from the background relative to the control in both cell lines were identified as hits to limit false positives. Standard ChIP-Seq experiments confirmed the presence of identified proteins that had no literature precedent for involvement in pluripotency. Knockdown of the TINC-enriched protein RCOR2 created cells that poorly differentiated from the stem cell state, demonstrating a previously unknown regulator of the pluripotency network. Several targets known to be involved in pluripotency identified in TINC samples were excluded due to trace presence in control samples, resulting in false negatives. The dual-TAL design to minimize false negatives prevented the direct application of label-free analysis packages to the mass spectrometry data, providing an opportunity for future development.

##### Genetic-Code Expansion Click-Chemistry TALs

Another strategy to enrich specific genomic loci utilizing TAL-based tools relies on the expansion of the genetic code [141]. Researchers systematically screened locations on a TAL to incorporate noncanonical amino acids to identify the optimal insertion site [142]. The noncanonical amino acid selected enables the modular TAL to be tagged with biotin through the use of bio-orthogonal click chemistry [143]. In the first proof-of-principle study, repetitive pericentromeric satellite III DNA was targeted. This region is the origin of nuclear stress bodies that form in response to conditions of heat stress. Negative controls demonstrated the chromatin region could only be isolated upon the addition of the bio-orthogonal biotin tag by the experimenter. This additional layer of chemical control to minimize false positives is analogous to TChP strategies. Subjecting both HeLa and HEK293 cells to heat shock conditions allowed for the enrichment of heat shock-associated proteins via label-free quantification of mass spectrometry data. Though several true positives were identified, the absence of heat shock factor 1 (HSF1) was surprising. Standard ChIP-qPCR experiments showed enrichment of HSF1 under heat shock conditions, indicating the protein was present at levels below the detection limit of the mass spectrometer after purification. However, other expected factors were identified in response to heat shock, indicating the ability of this tool to isolate locus-specific chromatin. This observation highlights an important consideration in reverse-ChIP experiments: though the techniques can often be reliable for detecting true locus-specific interaction partners, technology will need to be more advanced before it is possible to construct the entirety of a locus-specific proteome, including stoichiometry and posttranslational modifications.

#### 3.2.6. Epi-Decoder

An innovative report described a method to determine proteome composition at a specific genomic locus in yeast utilizing protein-based purification and a sequencing readout [144]. This technique, Epi-Decoder, relies on crossing yeast strains in which individual proteins are modified with epitope tags with strains in which the genomic region under study has unique DNA barcodes incorporated (Figure 4e) [145]. The pooled libraries are subjected to conventional ChIP-Seq with an antibody against the common epitope tag. Since each protein-tagged strain is associated with a unique barcode, sequencing counts of the barcode are a direct measure of protein occupancy at the locus. Since the signal can be amplified by PCR before sequencing, this technique does not require as much sample input as mass spectrometric techniques. Specific binding of over 70% of the yeast proteome was quantified in the initial report. Measurable differences in protein occupancy were detected both up- and downstream of the gene under study. Since the barcodes were only separated by 1.5 kb, this indicates a high degree of resolution for the technique. Epi-Decoder is a powerful tool for measuring changes in locus-specific proteomes in response to physiological cues cells receive [146]. Using a marker gene barcoded at the *HO* locus, the specific binding of approximately 700 chromatin proteins could be determined in triplicate within a single experimental run. This multiplexing capability enabled time-course measurements of protein occupancy at the locus in response to inhibition of transcription. Chemical inhibition of RNA polymerase led to rapid reduction in the binding of RNAPII accompanied by reduced binding of many core transcription proteins and gain of chromatin remodelers. The ability to measure local proteome composition in time course experiments is a powerful tool for studying regulation in response to cellular signals and points towards the future of reverse-ChIP experiments: locus-specific isolation of a chromatin region of interest during differentiation or in response to physiological perturbations.

### 3.3. CRISPR-Based Approaches

The discovery that the Cas9 endonuclease from the CRISPR system could be specifically targeted to an arbitrary genetic sequence via an engineered single guide RNA (sgRNA) has greatly expanded the realm of tractable experiments [147]. Beyond the explosion of use for genetic editing and engineering, a great deal of research has focused on the development of catalytically dead Cas9 (dCas9) as a tool for genomics [148]. A variety of strategies have employed the modular RNA-guided binding capabilities of dCas9 as a tool to enable the biochemical purification of specific chromatin regions for use in reverse-ChIP experiments (Figure 5) [149]. These dCas9-based approaches gain the advantages of nucleic acid-based targeting without succumbing to the inefficient purification that plagues PICh and related protocols or the laborious multi-step genetic engineering required for most protein-based methods.

#### 3.3.1. CRISPR-enChIP

The promise of dCas9 for use in the biochemical purification of chromatin was rapidly recognized. A CRISPR-based enChIP strategy studying the *IRF-1* locus in HEK293T cells was published concurrently with the approach utilizing TALs to study telomeres (vide supra) (Figure 5a) [150]. Proteins associated with the locus could be detected by MS after transient transfection of a FLAG-tagged dCas9 with a designed sgRNA. A subsequent report utilized a retroviral system for stable incorporation of dCas9 and used SILAC as a readout to quantitatively detect proteins that respond to IFNγ stimulation [151]. Novel candidate proteins found using the CRISPR-enChIP system were confirmed by conventional ChIP experiments to increase association with the *IRF-1* locus after stimulation. In a SILAC-based CRISPR-enChIP-MS approach, multiple proteins were found to change the association with the EPAS1 promoter under hypoxic conditions in neuroblastoma cells [152]. The transcription factor HDX significantly dissociated from the promoter under hypoxic conditions. Bioinformatic analysis revealed two putative binding sites for the protein 1–2 kb downstream of the transcription start site, giving a plausible explanation of the observed results. In a separate report from an independent group, researchers used a series of elegant CRISPR experiments to pinpoint a previously unidentified enhancer of the cartilage master transcription factor SOX9 1 Mbp upstream of the gene [153]. After identifying the enhancer and confirming its phenotypic output via the creation of enhancer deletion mice, researchers used CRISPR-enChIP with label-free mass spectrometry to identify Stat3 as the *trans*-acting factor binding to the cartilage-specific enhancer. CRISPR-enChIP has also been used by another research group to study the WNT5A promoter in triple-negative breast cancer via transient transfection of 3xFLAG-dCas9 and spectral counting mass spectrometry [154]. The data indicate that FOXC1 elicits noncanonical WNT5A signaling via the NF-kB pathway and is essential for the invasiveness of breast cancer. Researchers have also used a TMT-based CRISPR-enChIP approach to pull down the promoter of UCP1 in adipocytes in response to signaling by FGF6 [155]. The protein FLII and its binding partner LRRFIP1 increased association with the promoter, and the knockdown of LRRFIP1 significantly impaired the FGF6-induced UCP1 expression. These reports from several different labs in a variety of cell systems indicate the robustness of CRISPR-enChIP as a locus-specific chromatin purification strategy.

Some researchers have utilized CRISPR-enChIP to interrogate repetitive genomic elements beyond telomeres and centromeres. More than half the genome is composed of repetitive elements, and these regions are a large source of human genetic variation and have contributions to health and disease [156,157]. In addition to biological interest, these regions with multiple copies of elements of interest provide a technical advantage in reverse-ChIP approaches by increasing the signal that can be obtained from each cell based on the number of copies of the repetitive element. A CRISPR-enChIP approach studied the D4Z4 array containing multiple copies of the DUX4 gene in myoblasts [158]. This gene is expressed in early embryonic stages before being silenced in somatic tissues. Aberrant activation in skeletal muscle causes a form of muscular dystrophy. Mass spectrometry followed by functional studies demonstrated the NuRD and CAF-1 complexes repress the expression of DUX4 in myoblasts and iPS cells. A separate CRISPR-enChIP report studied Alu retrotransposon repeats surrounding the key pluripotency gene Nanog [159]. In this approach, cells were transiently transfected with 3xFLAG dCas9 with or without retinoic acid to induce differentiation. During differentiation, the aryl hydrocarbon receptor and CTCF were identified as key players interacting with the Alu elements to form a chromatin loop and repress Nanog.

#### 3.3.2. CRISPR-ChAP-MS

A PrA-tagged dCas9 was used to study the *GAL1* promoter in yeast in a CRISPR-ChAP-MS approach [160]. In contrast to TAL-ChAP-MS, the target region was enriched in both transcriptionally active and silent states, albeit with an order of magnitude greater enrichment for transcriptionally active chromatin. Label-free quantitative mass spectrometry in transcriptionally active conditions created a list of candidate interactor proteins, some of which were confirmed via conventional ChIP. Though the target region was enriched under repressive conditions, predicted off-target sites were also enriched to a comparable degree. This observation highlights the need to carefully design the sgRNA and perform control experiments to determine off-target binding and the impact of chromatin state on target locus enrichment. A subsequent study from an independent research group utilized CRISPR-ChAP-MS in two colon cancer lines with varying degrees of metastatic potential to study the promoter of MACC1, a metastasis-associated gene [161]. Researchers also utilized a biotinylated DNA probe representing 500 bp of the promoter region to identify potential regulatory proteins. A total of 24 proteins were identified by the CRISPR-ChAP-MS approach, and over 700 were associated with the biotinylated oligonucleotide probe. Conventional ChIP-PCR experiments demonstrated several of these candidates were bound to the targeted region. A subset of these identified proteins was uniquely identified in each cell line, indicating cell-type specific regulators that could be responsible for the differing metastatic potential between the cell lines.

#### 3.3.3. Cas9 Locus-Associated Proteome (CLASP), an In Vitro enChIP Approach

Purification of a genomic locus with dCas9 can be performed without expressing the protein in the cell type of interest, extending the experimental possibilities by removing the need for genetic engineering of the cell line of interest (Figure 5b). However, purification efficiency is typically lower, requiring a larger initial cell input (a difference of about 10-fold). Recombinant dCas9 ribonucleoproteins with an associated sgRNA can be used for in vitro enChIP [162]. This stands in contrast to attempts at in vitro enChIP using TALs, which were unsuccessful. This approach was used in a separate laboratory to study the histone cluster in *Drosophila melanogaster* S2 cells in a technique called CLASP (Cas9 locus-associated proteome) [163]. Label-free quantitation techniques were used to identify candidate regulatory proteins from the MS data. Knockdown of these genes impacted H2 expression, indicating their functional relevance. However, no classic sequence-specific transcription factors were observed. The authors posit two plausible explanations for this observation: the difference in time scales between the cross-linking step and the residence time of traditional transcription factors means they may not be captured; alternatively, the low abundance of TFs means that they might not be detected due to lack of sensitivity. Despite the absence of these traditional transcriptional regulators, CLASP successfully uncovered regulators of the targeted locus whose functional importance was orthogonally confirmed.

#### 3.3.4. CRISPR Affinity Purification In Situ of Regulatory Elements (CAPTURE)

The power of dCas9-based chromatin purification strategies was extended with the disclosure of the CRISPR affinity purification in situ of regulatory elements (CAPTURE) technique (Figure 5c) [164]. CAPTURE relies on the transduction of the cell line of interest with a dCas9 protein modified to include an acceptor site for biotinylation by the BirA biotin ligase. Streptavidin purification of biotinylated dCas9 successfully isolates the region targeted by the sgRNA with high efficiency. In comparison with dCas9 or FLAG antibody-based enrichment, biotin-streptavidin purification showed 18- or 284-fold on-target enrichment, respectively. The authors first identified known and novel telomere-associated proteins in proof-of-concept iTRAQ experiments. CAPTURE was then extended to target non-repetitive regions: iTRAQ experiments with dCas9 targeted to the β-globin cluster identified candidate proteins that were confirmed to interact with the locus via ChIP-Seq experiments. The functional relevance of the proteins in regulating the β-globin cluster was demonstrated with RNAi-mediated gene knockdown of identified factors. Subsequently, another research group utilized the CAPTURE system to study the *Nanog* promoter in mouse embryonic stem cells [165]. Many RNA binding proteins were identified, confirmed to bind to the region by ChIP-qPCR, and demonstrated to have an impact on self-renewal, pluripotency, and differentiation through knockdown experiments. Though these true positives were found, well-known factors, including OCT4 and SOX2 that are known to bind and regulate the promoter, were not identified. These techniques used 10^8^–10^9^ cells as input for proteomic experiments.

Sequence-defined chromatin purification by the CAPTURE technique can also reveal long-range DNA contacts by tying to a sequencing readout. These experiments are analogous to chromatin conformation capture (3C) [166] and related techniques that have been adopted widely to study three-dimensional genome architecture [167]. CAPTURE-3C-Seq allowed for the study of the behavior of super enhancers, using differentiation of embryonic stem cells as the model system [164]. Downregulation of genes regulated by the super enhancers during differentiation was correlated with a marked decrease in long-range DNA contacts made by these regulatory regions. Importantly, these changes in long-range interactions preceded any observable changes in chromatin accessibility or levels of H3K27ac, suggesting a specific order of changes to molecular chromatin architecture during cellular differentiation. A simplified CAPTURE system taking advantage of endogenous biotin protein ligases in eukaryotic cells demonstrated the ability to study many enhancers in a single experiment in a multiplexed fashion [168]. Utilizing endogenous biotin ligases precludes the need to engineer cells with BirA, reducing the number of laborious genetic engineering steps. The multiplexed capabilities of CAPTURE 2.0 enabled the classification of hierarchical structures within super enhancers as well as their temporal dynamics. Long-range DNA contacts have also been reported with CRISPR-enChIP-Seq [169].

#### 3.3.5. Proximity Labeling-Based CRISPR Approaches

Researchers have utilized proximity labeling to study chromatin in order to take advantage of the uniquely strong interaction between biotin and streptavidin [170]. These techniques covalently attach biotin to all biomolecules within a close physical radius (~10–20 nm) to the targeted region (Figure 5d) [171,172]. Directly biotinylating proteins associated with a chromatin locus via dCas9 targeting enables stringent washing protocols that could potentially remove indirect interaction partners using other CRISPR-based techniques. Furthermore, since the local proteome is directly tagged with a biotin handle for purification in live cells, there is no need for a cross-linking step. Multiple independent research groups have adopted proximity labeling-based strategies to isolate sequence-defined chromatin regions for analysis.

##### CasID

A BirA biotin ligase mutated to promiscuously attach biotin to all nearby lysine residues was fused to dCas9 for the CasID technique [173]. In the initial report, CasID was targeted to repetitive telomeric, major satellite, and minor satellite DNA in separate experiments to specifically label proteins contributing to the molecular chromatin architecture of these regions. After streptavidin purification and LC-MS, proteins were quantitatively identified using the label-free MaxLFQ technique. One-third of the proteins significantly enriched at major satellite repeats were also detected with the PICh technique, but CasID used an order of magnitude fewer cells (4 × 10^7^ cells cultured for 24 h). Newly identified proteins were confirmed to interact with the region targeted by the sgRNA via super-resolution microscopy in live cells, orthogonally confirming the results from the CasID experiments. In an extension of the CasID work by a separate research group, the protein desmoplakin, previously known only for its role in cell-cell adhesion, was identified as a telomere-associated protein and confirmed through functional experiments to play a role in maintaining telomere length [174]. Desmoplakin was not identified by other telomere-targeting purification methods, including PICh, indicating the utility of CasID to identify novel proteins specifically interacting with a targeted locus. Technical considerations for experiments and the development of CasID have been reviewed elsewhere [175].

##### CAPLOCUS

An alternative proximity labeling strategy leverages the engineered peroxidase APEX2 [176]. This approach generates highly reactive phenoxyl radicals that covalently link biotin to all surface-exposed tyrosine residues in a small radius within 1 min of labeling time. One dCas9-APEX2 approach extended the sgRNA to include two MS2 RNA elements. The MS2 coat protein that binds the structure formed by the RNA elements was fused to APEX2 to enable recruitment of the proximity labeling enzyme to the chosen genomic locus in a technique termed CAPLOCUS [177]. This technique could purify both repetitive telomeric and single-copy sequences with high specificity as confirmed through a sequencing readout. Interestingly, chromatin accessibility did not impact the performance of CAPLOCUS since target enrichment was not correlated with chromatin accessibility as measured by the ATAC-Seq assay [178]. CAPLOCUS was also used to determine telomere-associated RNA molecules and long-range DNA interactions with a sequencing readout; 4C-Seq confirmed many of the long-range interactions identified with the proximity labeling technique. A SILAC-based mass spectrometry protocol was used to identify telomere-associated proteins by utilizing telomere-targeting sgRNA as the experimental probe with a sgRNA targeting the non-telomere-associated Gal4 locus as a control for nonspecific, background labeling. Many of the identified telomere proteins were previously found in the literature, indicating the reliability of the CAPLOCUS technique. A subsequent study utilized CAPLOCUS to study proteins associated with the HBV mini-chromosome in human cells infected by the virus [179]. Several gRNAs were designed to tile the HBV mini-chromosome and sgGal4 was used as a negative control. The protein kinase PRKDC was the top hit identified. Subsequent experiments demonstrated PRKDC recruits RNAPII to the locus and that knockdown of the gene decreases HBV transcription, indicating it has potential as a therapeutic target.

##### dCas9-APEX2 Biotinylation at Genomic Elements by Restricted Spatial Tagging (C-BERST)

The direct fusion of APEX2 to dCas9 has also been utilized for genetically targeted proximity labeling. In a technique termed dCas9-APEX2 biotinylation at genomic elements by restricted spatial tagging (C-BERST), U2OS cells were engineered to contain a dCas9-APEX2 fusion protein under the control of a Tet-On promoter [180]. The construct also contained a degradation domain to enable more precise chemical control of expression similar to TChP [181]. Initial label-free quantitative mass spectrometry experiments with a sgRNA targeting telomeres vs. a nonspecific sgRNA not found in the human genome [182] demonstrated the specific enrichment of known telomere-associated proteins. Following these results, the authors used a triple SILAC approach with a telomere-targeting sgRNA cell line grown in heavy media, the nonspecific sgRNA cell line grown in medium media, and untransduced cells grown in light media. They were able to quantify many known telomere-targeting proteins, showing a good overlap with PICh and other telomere-targeting approaches. Live cell imaging confirmed the telomeric localization of a previously unknown partner. A separate triple SILAC experiment targeting centromeric α-satellite arrays showed good overlap with PICh and HyCCAPP experiments. 

##### Genetic Locus Proteomics (GLoPro)

In a concurrent report, a dCas9-APEX2 approach termed genetic locus proteomics (GLoPro) demonstrated the ability to target non-repetitive, single-copy loci [183]. In this report, the authors used sgRNAs targeting either the *hTERT* or *MYC* promoters in separate experiments. *hTERT* encodes the human telomerase protein whose function has been implicated in a multitude of cancers [184,185,186,187,188,189,190,191]. *MYC* encodes an oncogene commonly expressed in cancers [192,193,194,195,196].

Control GLoPro experiments showed a labeling radius of ~400 bp (~3 nucleosomes). Key to success for targeting single-copy loci was the tiling approach used for sgRNAs—multiple separate engineered sgRNA cell lines targeting the promoter region allowed samples to be pooled to increase signal. Additionally, any interactions disrupted by the binding of dCas9-APEX2 in any individual sgRNA cell line can be picked up by the other cell lines. A TMT-based quantitative proteomics approach used four to five individual sgRNA cell lines targeting each promoter along with a no-sgRNA control. Highly abundant proteins, such as histones and subunits of RNA polymerase, were identified but not enriched (presumably due to high background), but several classic sequence-specific transcription factors were enriched and confirmed to interact with each locus through conventional ChIP experiments. This observation highlights an important consideration in reverse-ChIP experiments: proteins with high levels of background expression are more difficult to identify as locus-specific binders. In other words, necessary attempts to minimize false positives can often result in false negatives.

The GLoPro technique was adapted to study the regulation of the *Ripk3* locus in mouse epithelial cells by a separate research group [197]. RIPK3 is a tumor suppressor gene involved in necroptosis that has been implicated as having a role in several cancers [198,199,200,201,202]. The oncogene MYC interacts with RIPK3 in the cytosol to prevent its tumor suppressor activity [203]. Three sgRNAs were designed to target the promoter region near the transcription start site of the gene. A nontargeting sgRNA was used as a negative control. More than 2000 proteins were identified in each cell line by GLoPro using the MaxQuant algorithm. Of those, nearly 200 were present in all three targeting sgRNA cell lines while being absent from the nontargeting line. Two proteins identified for follow-up studies were shown to bind directly to the *Ripk3* promoter by ChIP-qPCR. Knockdown of the factors also prevented the increase in RIPK3 transcription typically observed upon TNFα stimulation. Confirmation of putative transcriptional regulators by both ChIP and functional assays gives high confidence that GLoPro is a powerful tool for identifying novel contributors to locus-specific proteomes.

#### 3.3.6. Considerations for Experimental Design with dCas9

Several key considerations in designing a dCas9-based reverse-ChIP experiment emerge from the numerous reports leveraging the modular sgRNA-targeted CRISPR system. As with all reverse-ChIP approaches, control experiments are necessary to demonstrate that expression and/or binding of dCas9 does not disrupt the regulatory function of the region under study. Unique to dCas9-based experiments is the need to carefully design sgRNAs to minimize off-target binding and the associated control experiments to determine the degree of enrichment for predicted off-target sites. Minimizing expression levels of dCas9 is crucial when sgRNAs target non-repetitive regions. For single-copy loci, only two dCas9 molecules can specifically bind per diploid cell; increased expression of dCas9 provides more background, which necessitates larger input volumes to increase signal. Tiling sgRNAs to target single-copy loci can increase the relevant signal and identify proteins for which binding is disrupted in any individual sgRNA cell line. Appropriate negative controls must be included to filter out nonspecific interactions. Important negative controls include not expressing dCas9 to identify background binders, expression of dCas9 with no sgRNA or a nonspecific sgRNA to identify off-target effects, and exclusion of chemicals necessary to induce biotinylation in proximity labeling experiments. Using a nonspecific sgRNA provides a cleaner control than not expressing a sgRNA since the dynamics of dCas9 interrogating the genome varies with the presence or absence of a sgRNA. In some cases, a control sgRNA targeting a specific region far from the gene under study can be used (e.g., use of Gal4 as a control for telomere labeling as described above). Proximity labeling-based experiments also have special considerations. There is a differential abundance of surface-exposed lysine residues vs. surface-exposed tyrosine residues, so BirA vs. APEX2 experiments will have different biases for what proteins can be identified. Another important consideration is the difference in time scales for each method (hours for BirA vs. minutes or less for APEX2). Last, as with all reverse-ChIP approaches, any proteins identified must be confirmed to be functionally relevant via orthogonal experiments and/or confirmation from the literature.

## 4. Conclusions and Future Directions

Understanding the vast complexity of gene regulation necessitates tools that enable the unbiased quantitative identification of molecular factors interacting with any arbitrarily selected genomic locus. Many labs have devoted an enormous amount of effort to developing strategies for the selective purification of any chosen sequence-defined region. Though these efforts have uncovered novel biological insights, they are often only performed by specialist laboratories that have the expertise to effectively engage in labor-intensive processes. The emergence of dCas9-based reverse-ChIP strategies provides a modular tool that shows high potential for more widespread adoption. Increasingly powerful and sophisticated mass spectrometry protocols coupled with chromatin purification construct a list of candidate regulators that must be confirmed through orthogonal experiments. Advancements in mass spectrometry techniques will continue to provide opportunities to advance locus-specific chromatin isolation techniques (Box 1). Chromatin-associated RNAs and long-range DNA contacts can be identified via a sequencing readout to provide a more complete picture of chromatin architecture by capturing non-proteinaceous molecular regulators. Unbiased identification of the molecular components of transcriptional regulation at an arbitrarily chosen locus will enable researchers to build a mechanistic understanding of cellular processes ranging from differentiation to cell cycle progression to oncogenesis and metastasis.

Box 1Signal Boosting in Mass Spectrometry with Tandem Mass Tags (SCoPE-MS).A current frontier in quantitative mass spectrometry research is the development of techniques for single-cell proteomics [204,205]. The first report of single-cell proteomics by mass spectrometry (SCoPE-MS) took advantage of the multiplexing capabilities of TMTs [206]. Using a 10-plex TMT setup, researchers labeled eight individual cells with TMTs, then pooled them together with an empty TMT channel (i.e., no cells for negative control) and a signal boosting TMT channel in which hundreds of cells were labeled. The number of ions present during an LC run exceeds the ability of mass spectrometers to sample and identify every peptide present. The ions from the signal boosting channel provide enough signal for peptides to be identified, while the tandem mass tags enable the ratiometric comparison to quantitatively identify peptides in each individual cell. Improvements in sample processing [207] have allowed for characterization of cellular hierarchies using single cell proteomics [208] and for measurement of the proteome and transcriptome from individual cells [209]. Systematic characterization of settings for mass spectrometry runs has defined ratios of signal-boosting channels to experimental samples for single cell proteomics [210,211,212]. Utilization of simultaneous real-time search via synchronous precursor selection MS3 increases single cell proteome coverage [213]. All these strategies rely on the ratiometric comparison of results from trace samples with abundant samples that boost the signal for identification of peptides. Reverse-ChIP techniques require large sample inputs to produce enough signal for reliable detection (up to 10^9^–10^11^ cells in some protocols). The signal-boosting strategy from single cell proteomics provides a possible experimental paradigm to lessen the labor-intensive workflow in reverse-ChIP strategies. Individual cell lines used to isolate locus-specific chromatin could be labeled with TMTs using a smaller input volume, while the parent cell line can have ~10^9^ or more cells labeled to serve as the signal-boosting channel. The large number of proteins present in the parent cell line could serve as a platform for ‘amplification’ of the proteins present in the purified target region in an analogy to PCR.

Reverse-ChIP strategies have evolved from proof-of-concept experiments targeting repetitive genetic elements requiring specialist knowledge to strategies that allow for the study of single-copy loci by laboratories that have not devoted their careers to developing the necessary technology. These approaches have matured to the point that identification of functionally relevant molecular regulators at an arbitrarily selected locus of interest can be achieved. A longer-term goal will be to advance the techniques to the point that the entire proteome of a specific locus, including the stoichiometry of all components and identification of protein post-translational modifications, becomes routine. As adoption of reverse-ChIP protocols becomes more widespread, especially with the use of dCas9, the fields of transcriptional regulation and epigenetics can be expected to tackle long-standing questions with unprecedented molecular resolution. Development of an allele-specific reverse-ChIP strategy has the potential to provide a crucial tool to study genomic imprinting [214]. Expert labs that developed these technologies have already provided a roadmap for others to follow [215] and will continue to lead the way as they work to refine and extend the capabilities of the techniques. In the near future, it is possible that reverse-ChIP experiments will become as standardized and widely adopted as ChIP-Seq, becoming another tool to probe the interplay between molecular chromatin architecture and the functional output of the genome to understand the link between genotype and phenotype.

Adoption of reverse-ChIP techniques by cancer biologists will be invaluable in developing personalized cancer therapeutics. The phenotype of excessive cell growth embodied by cancer provides a technical advantage for contemporary reverse-ChIP experiments, which require large initial sample inputs. A future aim for clinical application of these technologies would be to create patient-specific, tumor-derived cell lines. Genomic and transcriptomic analysis can identify utilization of cancer specific promoters and other aspects of the cancer ‘regulome’ that can then be targeted for locus-specific chromatin purification. Identification of the molecular players controlling regulation at these key cancer genes in a patient-centered way will provide an avenue to develop personalized therapeutic regimens.

## Figures and Tables

**Figure 1 cells-12-01860-f001:**
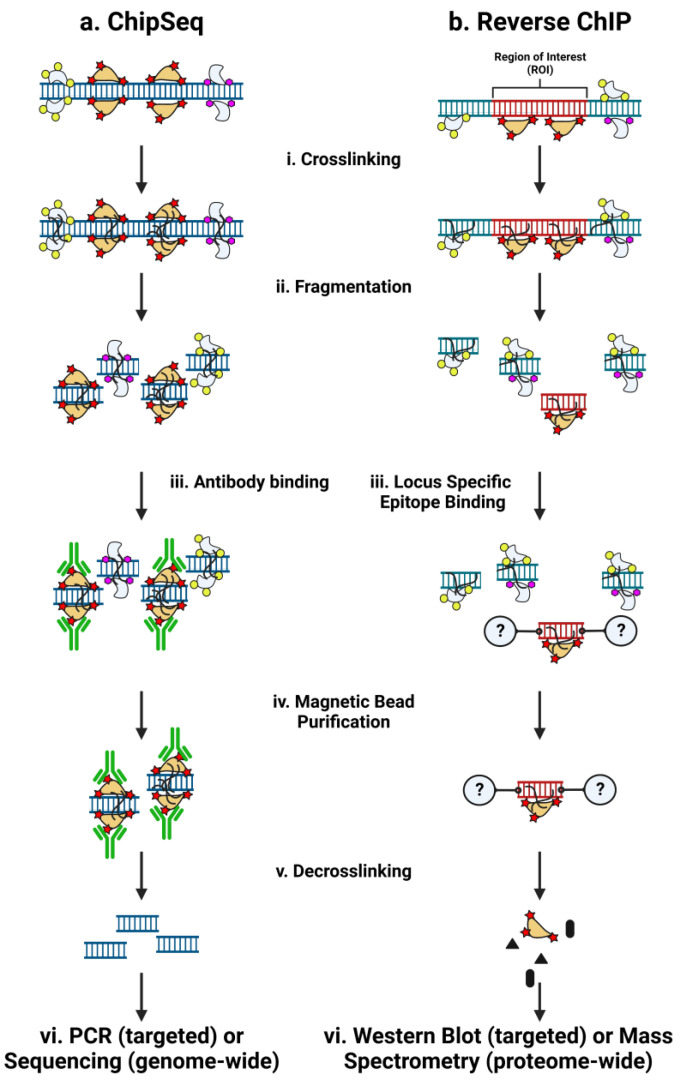
Comparison of ChIP-Seq with reverse-ChIP. ChIP-Seq (**a**) and reverse- ChIP (**b**) follow a similar experimental workflow. Cells are typically (though not always) cross-linked (i) so proteins and nucleic acids that comprise chromatin fragments are covalently bonded together. Subsequently, chromatin is fragmented (often through sonication) (ii) to solubilize chromatin complexes. The next step is epitope binding (iii). In ChIP-Seq, antibodies bind to the factor of interest. A variety of strategies have been employed to generate epitopes that bind to and isolate a specific genomic locus (see Section 3 of the review). The epitope-tagged region is purified via magnetic beads (iv) and molecules are de-cross-linked (v) for downstream analysis (vi). In ChIP-Seq, nucleic acids are isolated and identified through PCR (targeted) or sequencing (genome-wide). In Reverse-ChIP, the protein fraction is collected and identified via Western blot (targeted) or mass spectrometry (proteome-wide).

**Figure 2 cells-12-01860-f002:**
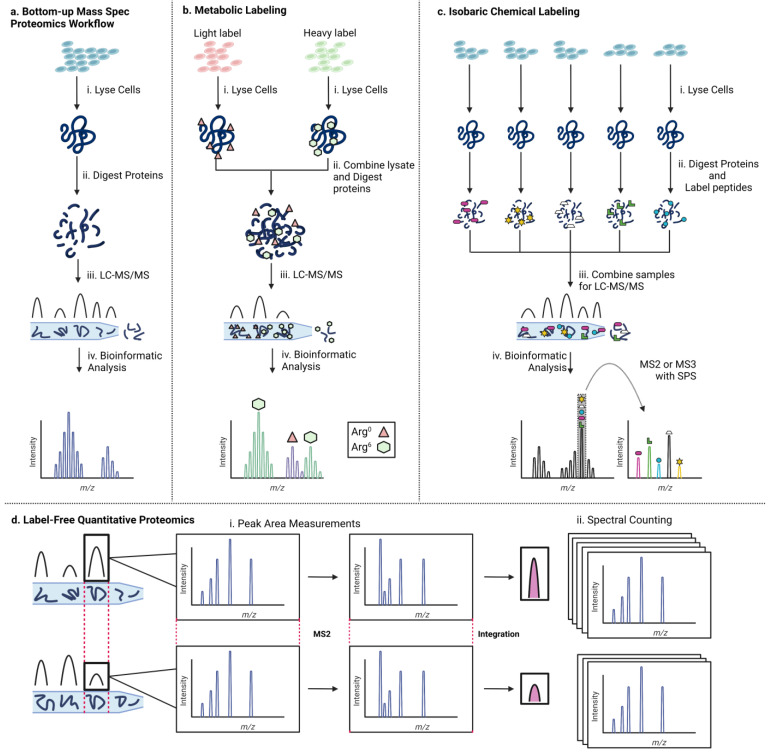
Approaches for quantitative mass spectrometry. (**a**) Workflow for a bottom-up mass spec proteomics experiment. Cells are lysed (i) and proteins are digested (ii). Peptide fragments are separated by physical properties using liquid chromatography coupled to mass spectrometry (iii), and bioinformatic analysis of produced spectra (iv) identifies peptides present in the sample. (**b**) Metabolic labeling for quantitative mass spectrometry. Cells in different conditions are grown on media that incorporate labeled amino acids of differing mass, which are incorporated into cells (e.g., Arg^0^, Arg^6^, or Arg^10^ [isotopically abundant Arg or Arg incorporating ^13^C_6_ and ^13^C_6_^15^N_4_, respectively]). Combined cell lysate is subjected to the conventional bottom-up proteomics workflow. Peptides present in both samples are doubled in *m/z*, while peptides uniquely present in one sample show a single signal. (**c**) Chemical labeling for quantitative mass spectrometry. Samples are processed according to the bottom-up proteomics workflow in parallel. Peptides are labeled with isobaric chemical tags before samples are combined for LC-MS. The mass of the tag is chosen for secondary fragmentation in MS/MS mode for ratiometric comparison of peptide quantity between multiplexed samples. To avoid ratio compression due to the co-isolation of contaminating peptides during MS2, an additional fragmentation step can be added for increased specificity (MS3, see Section 2.2.5). (**d**) Label-free approaches for quantitative mass spectrometry. Peptides are aligned across runs based on elution time on the LC, *m/z*, and spectra in MS and MS2 modes. Concentration is determined by the size of the peak in the chromatogram for peak area measurements as determined through integration (i). The number of times a particular *m/z* spectrum from a peptide is observed from a sample is used in spectral counting (ii).

**Figure 3 cells-12-01860-f003:**
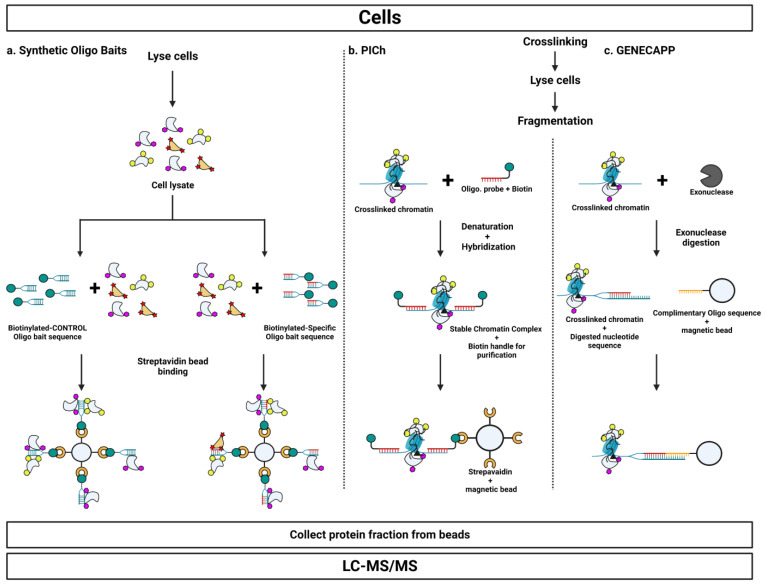
Nucleic acid-based approaches for locus-specific chromatin isolation. (**a**) Synthetic oligo baits can help identify proteins that recognize specific DNA sequences. Cells are lysed and split into control and sample pools. Biotinylated synthetic oligonucleotides with the bait sequence of interest or a scrambled control are added to the cell lysate to allow for proteins to bind. Sequence-specific interactors (orange with red stars) only interact with the bait sequence, while non-sequence-specific interactors bind to both bait and control. (**b**) Proteomics of isolated chromatin (PICh) and related protocols (HyCCAPP [yeast], RChIP [plants]) represent a conceptual shift in nucleic acid-based identification of sequence-specific binders by directly isolating the chromatin complexes from cells. After cross-linking and sonication to solubilize chromatin, a biotinylated oligonucelotide probe is added by the experimenter. DNA in chromatin complexes is denatured to allow for the oligo probe to form stable hybrids with the chromatin region of interest. Streptavidin-based purification isolates the chromatin region from the rest of the cellular milieu for downstream analysis. (**c**) Global exonuclease-based enrichment of chromatin-associated proteins for proteomics (GENECAPP) directly isolates chromatin complexes of interest similar to PICh via an alternative capture method. The addition of an exonuclease by the experimenter creates defined DNA overhangs that are targeted by magnetic bead-attached complementary oligonucleotides. This is in contrast to the denaturation-hybridization approach of PICh and related techniques. The chromatin complexes with the complementary oligo are purified via magnetic beads for downstream analysis.

**Figure 4 cells-12-01860-f004:**
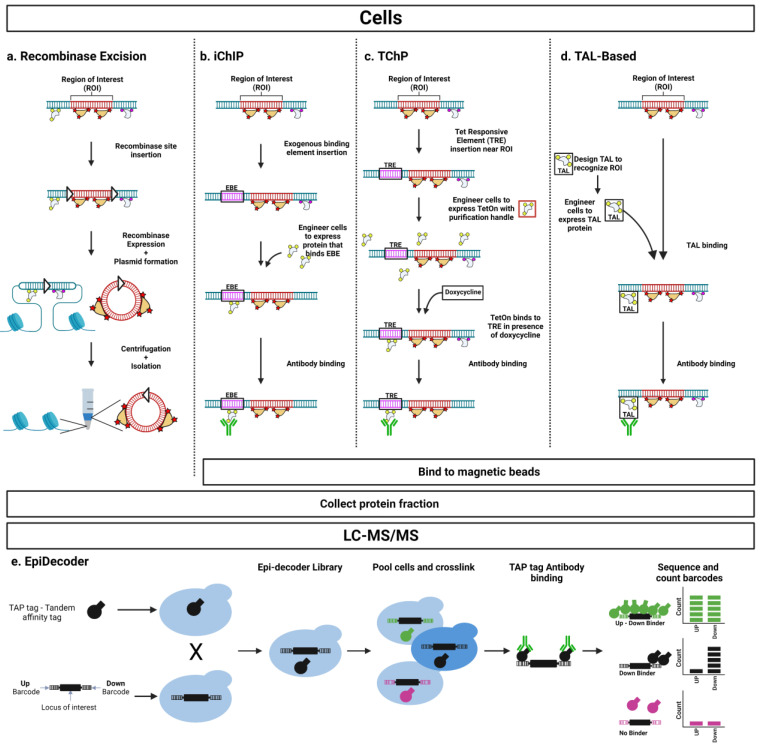
Protein-based purification strategies. (**a**) Recombinase sites are inserted to flank a region of interest (ROI) in recombinase excision. The cells are then engineered to express the recombinase that recognizes the introduced sequences to excise the region of interest as a small circular plasmid separate from bulk chromatin. Density centrifugation separates the ROI into the supernatant, and the protein fraction is collected for analysis. (**b**) For insertional chromatin immunoprecipitation (iChIP), an experimenter engineers cells to include an exogenous binding element (EBE) not natively present near the ROI (LexA-binding element in the initial report). The cells containing the EBE are then further engineered to express the protein that recognizes the EBE (LexA protein in the initial report). Since neither the EBE nor the protein that recognizes it is present in wild-type cells, antibody purification against the introduced protein can specifically isolate the ROI. (**c**) Targeted chromatin purification (TChP) follows a similar strategy to iChIP. The exogenous binding element introduced in TChP is the tet-responsive element (TRE). The cells are then engineered to express an affinity tag-modified TetOn protein, which only binds TRE in the presence of doxycycline. The inclusion or exclusion of doxycycline by the experimenter determines whether the specific locus or background nonspecific binders are isolated. (**d**) In TAL-based strategies, a transcription activator-like (TAL) protein is designed to specifically recognize and bind near the ROI. Cells are then engineered to express the TAL to enable purification. This protocol differs from iChIP and TChP in that the engineering of the recognition element is performed at the protein level and does not require the insertion of a recognition element near the ROI. (**e**) Epi-Decoder is a sequencing-based strategy in yeast. A library of yeast in which the ROI has been ‘barcoded’ with a unique molecular identifier up- and downstream is crossed with a library of yeast in which each individual protein has been tagged with a tandem affinity purification (TAP) tag. The result is a new library of yeast in which each protein is associated with a unique barcode. Conventional ChIP-Seq with an antibody directed at the common TAP tag allows for the identification of specific binding by any protein at the ROI based on sequencing counts of the unique barcode.

**Figure 5 cells-12-01860-f005:**
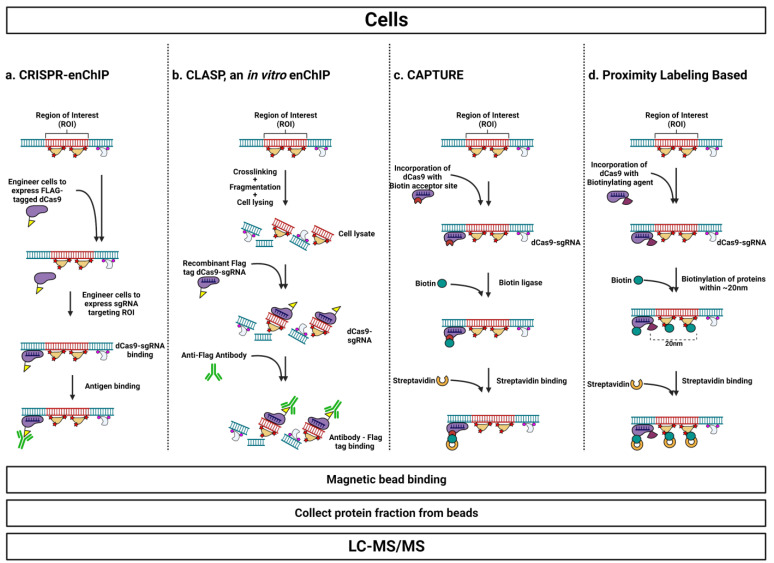
CRISPR-based purification approaches. (**a**) For CRISPR-enChIP and analogous strategies, an appropriate cell model system is engineered to incorporate an affinity-tagged dCas9 protein. This engineered cell line is then modified to express sgRNAs targeting the region of interest (ROI), allowing the dCas9-sgRNA complex to bind to chromatin. The ROI can then be purified with an antigen to the affinity tag on dCas9. The modular nature of sgRNA allows for relatively rapid targeting of new sites compared with more labor-intensive iChIP and TAL-based strategies. (**b**) The Cas9-locus-associated proteome (CLASP) technique is an in vitro version of CRISPR-enChIP strategies utilizing recombinant dCas9. Using recombinant dCas9 allows for the isolation of the ROI without needing to engineer the cell line to express the dCas9 protein. This stands in contrast to TAL-based strategies where recombinant TAL expression is not sufficient for locus isolation. However, purification efficiency is lower. Empirically, about an order of magnitude more cells are needed for in vitro strategies using recombinant protein. (**c**) For the CRISPR affinity purification in situ of regulatory elements (CAPTURE) technique, cells are engineered to express a dCas9 protein that has been modified to incorporate a site for in vivo biotinylation by a biotin ligase. The first-generation CAPTURE required engineering cells to express the biotin ligase as well, though the second-generation version takes advantage of endogenous biotin ligases. The biotinylated-dCas9-chromatin complex can then be purified using streptavidin beads for analysis of locus-specific binders. (**d**) Proximity labeling approaches rely on engineering cells to express dCas9 fused to proximity labeling enzymes, such as BirA or APEX2. The region of interest is targeted with designed sgRNAs, so the proximity labeling enzyme can covalently link biotin to proteins in the ROI. The spatial and temporal resolution of tagging depends on the specific proximity labeling enzyme used. Since proteins present at the locus are directly tagged with biotin in live cells, there is no need for a cross-linking step to covalently bind chromatin complexes together.

**Table 1 cells-12-01860-t001:** Techniques for isolating locus-specific chromatin organized by biochemical approach.

Purification Handle	Strategy (Variations)	Cross-Linking?	Genetic Engineering?	Notes
Nucleic Acid-Based	Synthetic Oligo	No	No	High background—only three proteins with >2-fold enrichment out of over 900 identified in the early report
	PICh (HyCCAPP [originally developed in yeast], qPICh, ePICh, RChIP [developed in plants])	Yes	No	Large sample input requirement—as many as 50 15-cm^2^ plates per locus in human cells.
	GENECAPP	Yes	No	Developed in vitro; only demonstrated in bacteria in vivo
Protein-Based	Recombinase Excision	Yes	Yes	Density centrifugation of excised chromosome to get a region of interest into the supernatant
	iChIP (ChAP-MS [originally developed in yeast])	Yes	Yes (two steps)	Region of interest can be isolated with recombinant protein purification
	TChP	Yes	Yes (two steps)	Binding of the purification handle is under chemical control
	TAL-Based (enChIP, TAL-ChAP-MS, TINC, click chemistry TALS)	Yes	Yes (one step)	Need to engineer a TAL that recognizes region of interest; recombinant TAL expression unsuccessful at purification
	EpiDecoder	Yes	Yes	Only in yeast; sequencing-based readout
CRISPR-Based	CRISPR-enChIP (CRISPR-ChAP-MS)	Yes	Yes	Adaptation of TAL strategies with dCas9 as targeting protein
	CLASP (in vitro CRISPR-enChIP)	Yes	No	CRISPR-enChIP using recombinant dCas9 expression
	CAPTURE	Yes	Yes	Site for biotinylation on dCas9 to allow for streptavidin-based purification
	Proximity labeling based (CasID, CAPLOCUS, C-BERST, GLoPro)	No	Yes	Differential biases for proximity labeling enzymes. BirA biotinylate lysine residues, APEX2 hits tyrosine residues

## Data Availability

Not applicable.

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
