# Peer review of "Reverse-ChIP Techniques for Identifying Locus-Specific Proteomes: A Key Tool in Unlocking the Cancer Regulome"

_cells, 2023, doi:10.3390/cells12141860_

Round 1

Reviewer 1 Report

The stated purpose of this review is to describe methods by which proteins bind at or near a specific locus in the human genome can be identified, acknowledging the usefulness of such methods in understanding the aberrant transcriptional regulation that occurs in cancers.  A review with that particular focus would be greatly appreciated by the scientific community. However, the majority of the review by MacKenzie et al. describes methods that have only really been successful when using lower organisms (e.g. yeast) or when studying repetitive regions (such as telomeres or repeat elements).  I think that this review would be most useful if it was refocused on only those methods that had been shown to work to identify proteins bound to single copy regions in human/mouse genomes under in vivo conditions. By reducing the descriptions of other methods, the authors could provide more information on the “more useful” techniques, such as how many cells are needed, has anyone other than the lab that published the first successful attempt gotten the technique to work, how many engineering steps are needed to perform the technique, etc.  With the advent of the dCas9 -related methods, it is likely that many of the “older” techniques will not be used by the field. More details describing pitfalls and requirements of the newer methods would be useful.

Reviewer 2 Report

The is a deep and well written Review from MacKenzie, Cisneros, Maynard, and Snyder. Overall, I enjoyed reading the review and found it carefully written. I am sure the readers of Cells will find this piece informative. Despite these strengths I have identified a number of minor suggestions that should improve the accuracy and readability.

Line 17: suggest “at loci of interest” for “arbitrary selected”.

Figure 1b bottom suggest “proteome-wide” for “genome-wide” for MS.

Line 112: unclear meaning “making unique identifications difficult” please articulate further.  

Line 125: suggest “properties” for “behavior”.

Line 168: suggest “optimizing” for “solving” and consider adding “reducing the non-specific background”, don’t love the use of “issues” here.

Line 176: I love MS and sure wish it was unbiased but unfortunately it does have some biases like requiring Ks and Rs, peptides have to fall M/Z scan range etc, suggest “discovery-based” for “unbiased manner”.

Line 197: suggest swapping “proteolytically digested” for “proteolytically degraded”, I don’t think digesting proteins to peptides reduced complexity per say. It does concentrate signals though. This sentence could be clarified.

Line 199: sentence starting with “A common technique..” this needs to be moved up, GelC is done on the protein level before in gel digestion.

Line 203: suggest “facilitate” for “help”.

Line 206: after present consider inserting “in a small m/z range” for DIA. Suggest swapping “database search” for “bioinformatic tools”.

Figure 2B should indicate isotopes. Figure 2C needs to have “isobaric” could also have MS3 and multinotch as well. Figure 2D needs to have MS1 and MS2 indicated, should also mention peak integration.

Section 2.1 should mention “isotopes” when discussing “heavy atoms”. Could also spell out the commonly and uncommonly used AA with special media lacking these AAs etc.

Section 2.1.1 include mention of heavy labeled Arg and Lys.   

Section 2.2.3 need to mention isobaric and explain the concept, power of MS3, multinotch and even the fact that now on the fly search with TMT is possible.

Section 2.3.1. Need include MS1 and peak integration.

Line 419 – 420, the benefits of SILAC versus TMT should be defined… TMT allows quant of same peptides across all samples but ratio compression, But… labeling is at the peptide level and SILAC allows for sample mixing before biochemistry.

I find the writing on section 3 clear and compelling. I also like the conclusion.

Round 2

Reviewer 1 Report

The revised version is now ready for acceptance.